# Learning Recurrent Binary/Ternary Weights

**Arash Ardakani, Zhengyun Ji, Sean C. Smithson, Brett H. Meyer & Warren J. Gross**
Department of Electrical and Computer Engineering, McGill University, Montreal, Canada
`{arash.ardakani, zhengyun.ji, sean.smithson}@mail.mcgill.ca`
`{brett.meyer, warren.gross}@mcgill.ca`

## Abstract

Recurrent neural networks (RNNs) have shown excellent performance in processing sequence data. However, they are both complex and memory intensive due to their recursive nature. These limitations make RNNs difficult to embed on mobile devices requiring real-time processes with limited hardware resources. To address the above issues, we introduce a method that can learn binary and ternary weights during the training phase to facilitate hardware implementations of RNNs. As a result, using this approach replaces all multiply-accumulate operations by simple accumulations, bringing significant benefits to custom hardware in terms of silicon area and power consumption. On the software side, we evaluate the performance (in terms of accuracy) of our method using long short-term memories (LSTMs) and gated recurrent units (GRUs) on various sequential models including sequence classification and language modeling. We demonstrate that our method achieves competitive results on the aforementioned tasks while using binary/ternary weights during the runtime. On the hardware side, we present custom hardware for accelerating the recurrent computations of LSTMs with binary/ternary weights. Ultimately, we show that LSTMs with binary/ternary weights can achieve up to $12\times$ memory saving and $10\times$ inference speedup compared to the full-precision hardware implementation design.

## 1 Introduction

Convolutional neural networks (CNNs) have surpassed human-level accuracy in various complex tasks by obtaining a hierarchical representation with increasing levels of abstraction (Bengio (2009); Lecun et al. (2015)). As a result, they have been adopted in many applications for learning hierarchical representation of spatial data. CNNs are constructed by stacking multiple convolutional layers often followed by fully-connected layers (Lecun et al. (1998)). While the vast majority of network parameters (i.e. weights) are usually found in fully-connected layers, the computational complexity of CNNs is dominated by the multiply-accumulate operations required by convolutional layers (Yang et al. (2015)). Recurrent neural networks (RNNs), on the other hand, have shown remarkable success in modeling temporal data (Mikolov et al. (2010); Graves (2013); Cho et al. (2014a); Sutskever et al. (2014); Vinyals et al. (2014)). Similar to CNNs, RNNs are typically over-parameterized since they build on high-dimensional input/output/state vectors and suffer from high computational complexity due to their recursive nature (Xu et al. (2018); Han et al. (2015)). As a result, the aforementioned limitations make the deployment of CNNs and RNNs difficult on mobile devices that require real-time inference processes with limited hardware resources.

Several techniques have been introduced in literature to address the above issues. In (Sainath et al. (2013); Jaderberg et al. (2014); Lebedev et al. (2014); Tai et al. (2015)), it was shown that the weight matrix can be approximated using a lower rank matrix. In (Liu et al. (2015); Han et al. (2015); Wen et al. (2016); Ardakani et al. (2016)), it was shown that a significant number of parameters in DNNs are noncontributory and can be pruned without any performance degradation in the final accuracy performance. Finally, quantization approaches were introduced in (Courbariaux et al. (2015); Lin et al. (2015); Courbariaux & Bengio (2016); Kim & Smaragdis (2016); Hubara et al. (2016b); Rastegari et al. (2016); Hubara et al. (2016a); Zhou et al. (2016); Li & Liu (2016); Zhu et al. (2016)) to reduce the required bitwidth of weights/activations. In this way, power-hungry multiply-accumulate operations are replaced by simple accumulations while also reducing the number of memory accesses to the off-chip memory.

Considering the improvement factor of each of the above approaches in terms of energy and power reductions, quantization has proven to be the most beneficial for hardware implementations. However, all of the aforementioned quantization approaches focused on optimizing CNNs or fully-connected networks only. As a result, despite the remarkable success of RNNs in processing sequential data, RNNs have received the least attention for hardware implementations, when compared to CNNs and fully-connected networks. In fact, the recursive nature of RNNs makes their quantization difficult. In (Hou et al. (2016)), for example, it was shown that the well-known *BinaryConnect* technique fails to binarize the parameters of RNNs due to the exploding gradient problem (Courbariaux et al. (2015)). As a result, a binarized RNN was introduced in (Hou et al. (2016)), with promising results on simple tasks and datasets. However it does not generalize well on tasks requiring large inputs/outputs (Xu et al. (2018)). In (Xu et al. (2018); Hubara et al. (2016b)), multi-bit quantized RNNs were introduced. These works managed to match their accuracy performance with their full-precision counterparts while using up to 4 bits for data representations.

In this paper, we propose a method that learns recurrent binary and ternary weights in RNNs during the training phase and eliminates the need for full-precision multiplications during the inference time. In this way, all weights are constrained to $\{+1, -1\}$ or $\{+1, 0, -1\}$ in binary or ternary representations, respectively. Using the proposed approach, RNNs with binary and ternary weights can achieve the performance accuracy of their full-precision counterparts. In summary, this paper makes the following contributions:

- We introduce a method for learning recurrent binary and ternary weights during both forward and backward propagation phases, reducing both the computation time and memory footprint required to store the extracted weights during the inference.

- We perform a set of experiments on various sequential tasks, such as sequence classification, language modeling, and reading comprehension. We then demonstrate that our binary/ternary models can achieve near state-of-the-art results with greatly reduced computational complexity.[1]

- We present custom hardware to accelerate the recurrent computations of RNNs with binary or ternary weights. The proposed dedicated accelerator can save up to $12\times$ of memory elements/bandwidth and speed up the recurrent computations by up to $10\times$ when performing the inference computations.

## 2 RELATED WORK

During the binarization process, each element of the full-precision weight matrix $\mathbf{W} \in \mathbb{R}^{d_I \times d_J}$ with entries $w_{i,j}$ is binarized by $w_{i,j} = \alpha_{i,j} w_{i,j}^B$, where $\alpha_{i,j} \geq 0$, $i \in \{1, \ldots, d_I\}$, $j \in \{1, \ldots, d_J\}$ and $w_{i,j}^B \in \{-1, +1\}$. In BinaryConnect (Courbariaux et al. (2015)), the binarized weight element $w_{i,j}^B$ is obtained by the sign function while using a fixed scaling factor $\alpha$ for all the elements: $w_{i,j}^B = \alpha \times \text{sign}(w_{i,j})$. In TernaryConnect (Lin et al. (2015)), values hesitating to be either +1 or -1 are clamped to zero to reduce the accuracy loss of binarization: $w_{i,j} = \alpha_{i,j} w_{i,j}^T$ where $w_{i,j}^T \in \{-1, 0, +1\}$. To further improve the precision accuracy, TernaryConnect stochastically assigns ternary values to the weight elements by performing $w_{i,j} = \alpha \times \text{Bernoulli}(|w_{i,j}|) \times \text{sign}(w_{i,j})$ while using a fixed scaling factor $\alpha$ for each layer. Ternary weight networks (TWNs) were then proposed to learn the factor $\alpha$ by minimizing the L2 distance between the full-precision and ternary weights for each layer. Zhou et al. (2016) introduced DoReFa-Net as a method that can learn different bitwidths for weights, activations and gradients. Since the quantization functions used in the above works are not differentiable, the derivative of the loss $\ell$ w.r.t the full-precision $\mathbf{W}$ is approximated by

$$\frac{\partial \ell}{\partial \mathbf{W}} \approx \frac{\partial \ell}{\partial \mathbf{W}^B} \approx \frac{\partial \ell}{\partial \mathbf{W}^T}, \tag{1}$$

where $\mathbf{W}^B$ and $\mathbf{W}^T$ denote binarized and ternarized weights, respectively.

Zhu et al. (2016) introduced the trained ternary quantization (TTQ) method that uses two assymetric scaling parameters ($\alpha_1$ for positive values and $\alpha_2$ for negative values) to ternarize the weights. In loss-

---

[1]The codes for these tasks are available online at `https://github.com/arashardakani/Learning-Recurrent-Binary-Ternary-Weights`

aware binarization (LAB) (Hou et al. (2016)), the loss of binarization was explicitly considered. More precisely, the loss w.r.t the binarized weights is minimized using the proximal Newton algorithm. Hou & Kwok (2018) extended LAB to support different bitwidths for the weights. This method is called loss-aware quantization (LAQ). Recently, Guo et al. (2017) introduced a new method that builds the full-precision weight matrix $\mathbf{W}$ as $k$ multiple binary weight matrices: $w_{i,j} \approx \sum_{z=1}^{k} \alpha_{i,j}^{k} \beta_{i,j}^{k}$ where $\beta_{i,j}^{k} \in \{-1, +1\}$ and $\alpha_{i,j}^{k} > 0$. Xu et al. (2018) also uses a binary search tree to efficiently derive the binary codes $\beta_{i,j}^{k}$, improving the prediction accuracy. While using multiple binary weight matrices reduces the bitwidth by a factor of $32\times$ compared to its full-precision counterpart, it increases not only the number of parameters but also the number of operations by a factor of $k$ (Xu et al. (2018)).

Among the aforementioned methods, only works of Xu et al. (2018) and Hou & Kwok (2018) targeted RNNs to reduce their computational complexity and outperformed all the aforementioned methods in terms of the prediction accuracy. However, they have shown promising results only on specific temporal tasks: the former targeted only the character-level language modeling task on small datasets while the latter performs the word-level language modeling task and matches the performance of the full-precision model when using $k = 4$. Therefore, there are no binary models that can match the performance of the full-precision model on the word-level language modeling task. More generally, there are no binary/ternary models that can perform different temporal tasks while achieving similar prediction accuracy to its full-precision counterpart is missing in literature.

## 3 PRELIMINARIES

Despite the remarkable success of RNNs in processing variable-length sequences, they suffer from the exploding gradient problem that occurs when learning long-term dependencies (Bengio et al. (1994); Pascanu et al. (2013)). Therefore, various RNN architectures such as Long Short-Term Memory (LSTM) (Hochreiter & Schmidhuber (1997)) and Gated Recurrent Unit (GRU) (Cho et al. (2014b)) were introduced in literature to mitigate the exploding gradient problem. In this paper, we mainly focus on the LSTM architecture to learn recurrent binary/ternary weights due to their prevalent use in both academia and industry. The recurrent transition of LSTM is obtained by:

$$
\begin{aligned}
\mathbf{f}_t &= \sigma\left(\mathbf{W}_{fh}\mathbf{h}_{t-1} + \mathbf{W}_{fx}\mathbf{x}_t + \mathbf{b}_f\right), \\
\mathbf{i}_t &= \sigma(\mathbf{W}_{ih}\mathbf{h}_{t-1} + \mathbf{W}_{ix}\mathbf{x}_t + \mathbf{b}_i), \\
\mathbf{o}_t &= \sigma(\mathbf{W}_{oh}\mathbf{h}_{t-1} + \mathbf{W}_{ox}\mathbf{x}_t + \mathbf{b}_o), \\
\mathbf{g}_t &= \tanh(\mathbf{W}_{gh}\mathbf{h}_{t-1} + \mathbf{W}_{gx}\mathbf{x}_t + \mathbf{b}_g), \\
\mathbf{c}_t &= \mathbf{f}_t \odot \mathbf{c}_{t-1} + \mathbf{i}_t \odot \mathbf{g}_t, \\
\mathbf{h}_t &= \mathbf{o}_t \odot \tanh(\mathbf{c}_t),
\end{aligned}
\tag{2}
$$

where $\{\mathbf{W}_{fh}, \mathbf{W}_{ih}, \mathbf{W}_{oh}, \mathbf{W}_{gh}\} \in \mathbb{R}^{d_h \times d_h}$, $\{\mathbf{W}_{fx}, \mathbf{W}_{ix}, \mathbf{W}_{ox}, \mathbf{W}_{gx}\} \in \mathbb{R}^{d_x \times d_h}$ and $\{\mathbf{b}_f, \mathbf{b}_i, \mathbf{b}_o, \mathbf{b}_g\} \in \mathbb{R}^{d_h}$ denote the recurrent weights and bias. The parameters $\mathbf{h} \in \mathbb{R}^{d_h}$ and $\mathbf{c} \in \mathbb{R}^{d_h}$ are hidden states. The logistic sigmoid function and Hadamard product are denoted as $\sigma$ and $\odot$, respectively. The updates of the LSTM parameters are regulated through a set of gates: $\mathbf{f}_t$, $\mathbf{i}_t$, $\mathbf{o}_t$ and $\mathbf{g}_t$. Eq. (2) shows that the main computational core of LSTM is dominated by the matrix multiplications. The recurrent weight matrices $\mathbf{W}_{fh}, \mathbf{W}_{ih}, \mathbf{W}_{oh}, \mathbf{W}_{gh}, \mathbf{W}_{fx}, \mathbf{W}_{ix}, \mathbf{W}_{ox}$ and $\mathbf{W}_{gx}$ also contain the majority of the model parameters. As such, we aim to compensate the computational complexity of the LSTM cell and reduce the number of memory accesses to the energy/power-hungry DRAM by binarizing or ternarizing the recurrent weights.

## 4 LEARNING RECURRENT BINARY/TERNARY WEIGHTS

Hou et al. (2016) showed that the methods ignoring the loss of the binarization process fail to binarize the weights in LSTMs despite their remarkable performance on CNNs and fully-connected networks. In BinaryConnect as an example, the weights are binarized during the forward computations by thresholding while Eq. (1) is used to estimate the loss w.r.t. the full-precision weights without considering the quantization loss. When the training is over, both the full-precision and binarized

weights can be then used to perform the inference computations of CNNs and fully-connected networks (Lin et al. (2015)). However, using the aforementioned binarization approach in vanilla LSTMs fails to perform sequential tasks due to the gradient vanishing problem as discussed in (Hou et al. (2016)). To further explore the cause of this problem, we performed a set of experiments. We first measured the probability density of the gates and hidden states of a binarized LSTM and observed that the binarized LSTM fail to control the flow of information (see Appendix A for more details). More specifically, the input gate $\mathbf{i}$ and the output gate $\mathbf{o}$ tend to let all information through, the gate $\mathbf{g}$ tends to block all information, and the forget gate $f$ cannot decide to let which information through. In the second experiment, we measured the probability density of the input gate $\mathbf{i}$ before its non-linear function applied (i.e., $\mathbf{i}^p = \mathbf{W}_{ih}\mathbf{h}_{t-1} + \mathbf{W}_{ix}\mathbf{x}_t + \mathbf{b}_i$) at different iterations during the training process. In this experiment, we observed that the binarization process changes the probability density of the gates and hidden states during the training process, resulting in all positive values for $\mathbf{i}^p$ and values centered around 1 for the input gate $\mathbf{i}$ (see Appendix A for more details). To address the above issue, we propose the use of batch normalization in order to learn binarized/ternarized recurrent weights.

It is well-known that a network trained using batch normalization is less susceptible to different settings of hyperparameters and changes in the distribution of the inputs to the model (Ioffe & Szegedy (2015)). The batch normalization transform can be formulated as follows:

$$BN(\mathbf{x}; \phi, \gamma) = \gamma + \phi \odot \frac{\mathbf{x} - \mathbb{E}(\mathbf{x})}{\sqrt{\mathbb{V}(\mathbf{x}) + \epsilon}}, \tag{3}$$

where $\mathbf{x}$ and $\epsilon$ denote the unnormalized vector and a regularization hyperparameter. The mean and standard deviation of the normalized vector are determined by model parameters $\phi$ and $\gamma$. The statistics $\mathbb{E}(\mathbf{x})$ and $\mathbb{V}(\mathbf{x})$ also denote the estimations of the mean and variance of the unnormalized vector for the current minibatch, respectively. Batch normalization is commonly applied to a layer where changing its parameters affects the distributions of the inputs to the next layer. This occurs frequently in RNN where its input at time $t$ depends on its output at time $t - 1$. Several works have investigated batch normalization in RNNs (Cooijmans et al. (2016); Laurent et al. (2016); Amodei et al. (2016)) to improve their convergence speed and performance.

The main goal of our method is to represent each element of the full-precision weight $\mathbf{W}$ either as $w_{i,j} = \alpha w_{i,j}^B$ or $w_{i,j} = \alpha w_{i,j}^T$, where $\alpha$ is a fixed scaling factor for all the weights and initialized from Glorot & Bengio (2010). To this end, we first divide each weight by the factor $\alpha$ to normalize the weights. We then compute the probability of getting binary or ternary values for each element of the full-precision matrix $\mathbf{W}$ by

$$P(w_{i,j} = 1) = \frac{w_{i,j}^N + 1}{2}, P(w_{i,j} = -1) = 1 - P(w_{i,j} = 1), \tag{4}$$

for binarization and

$$P(w_{i,j} = 1) = P(w_{i,j} = -1) = |w_{i,j}^N|, P(w_{i,j} = 0) = 1 - P(w_{i,j} = 1), \tag{5}$$

for ternarization, where $w_{i,j}^N$ denotes the normalized weight. Afterwards, we stochastically sample from the Bernoulli distribution to obtain binarized/ternarized weights as follows

$$w_{i,j}^B = \text{Bernoulli}(P(w_{i,j} = 1)) \times 2 - 1, w_{i,j}^T = \text{Bernoulli}(P(w_{i,j} = 1)) \times \text{sign}(w_{i,j}). \tag{6}$$

Finally, we batch normalize the vector-matrix multiplications between the input and hidden state vectors with the binarized/ternarized weights $\mathbf{W}_{fh}^{B/T}, \mathbf{W}_{ih}^{B/T}, \mathbf{W}_{oh}^{B/T}, \mathbf{W}_{gh}^{B/T}, \mathbf{W}_{fx}^{B/T}, \mathbf{W}_{ix}^{B/T}, \mathbf{W}_{ox}^{B/T}$ and $\mathbf{W}_{gx}^{B/T}$. More precisely, we perform the recurrent computations as

$$\mathbf{f}_t = \sigma\left(BN(\mathbf{W}_{fh}^{B/T}\mathbf{h}_{t-1}; \phi_{fh}, 0) + BN(\mathbf{W}_{fx}^{B/T}\mathbf{x}_t; \phi_{fx}, 0) + \mathbf{b}_f\right),$$

$$\mathbf{i}_t = \sigma\left(BN(\mathbf{W}_{ih}^{B/T}\mathbf{h}_{t-1}; \phi_{ih}, 0) + BN(\mathbf{W}_{ix}^{B/T}\mathbf{x}_t; \phi_{ix}, 0) + \mathbf{b}_i\right),$$

$$\mathbf{o}_t = \sigma\left(BN(\mathbf{W}_{oh}^{B/T}\mathbf{h}_{t-1}; \phi_{oh}, 0) + BN(\mathbf{W}_{ox}^{B/T}\mathbf{x}_t; \phi_{ox}, 0) + \mathbf{b}_o\right),$$

$$\mathbf{g}_t = \tanh\left(BN(\mathbf{W}_{gh}^{B/T}\mathbf{h}_{t-1}; \phi_{gh}, 0) + BN(\mathbf{W}_{gx}^{B/T}\mathbf{x}_t; \phi_{gx}, 0) + \mathbf{b}_g\right). \tag{7}$$

Table 1: Testing character-level BPC values of quantized LSTM models and size of their weight matrices in terms of KByte.

| Model | Precision | Linux Kernel | | War & Peace | | Penn Treebank | |
| --- | --- | --- | --- | --- | --- | --- | --- |
| | | Test | Size | Test | Size | Test | Size |
| LSTM (baseline) | Full-precision | 1.73 | 5024 | 1.72 | 4864 | 1.39 | 16800 |
| LSTM with binary weights (ours) | Binary | **1.79** | 157 | **1.78** | 152 | **1.43** | 525 |
| BinaryConnect (Courbariaux et al. (2015)) | Binary | 4.24 | 157 | 5.10 | 152 | 2.51 | 525 |
| LAB (Hou et al. (2016)) | Binary | 1.88 | 157 | 1.86 | 152 | 1.56 | 525 |
| LSTM with ternary weights (ours) | Ternary | **1.75** | 314 | **1.72** | 304 | **1.39** | 1050 |
| TWN (Li & Liu (2016)) | Ternary | 1.85 | 314 | 1.86 | 304 | 1.51 | 1050 |
| TTQ (Zhu et al. (2016)) | Ternary | 1.88 | 314 | 1.83 | 304 | 1.49 | 1050 |
| LAQ (Hou & Kwok (2018)) | Ternary | 1.81 | 314 | 1.80 | 304 | 1.46 | 1050 |
| LAQ (Hou & Kwok (2018)) | 3 bits | 1.84 | 471 | 1.83 | 456 | 1.46 | 1575 |
| LAQ (Hou & Kwok (2018)) | 4 bits | 1.90 | 628 | 1.83 | 608 | 1.47 | 2100 |
| DoReFa-Net (Zhou et al. (2016)) | 3 bits | 1.84 | 471 | 1.95 | 456 | 1.47 | 1575 |
| DoReFa-Net (Zhou et al. (2016)) | 4 bits | 1.90 | 628 | 1.92 | 608 | 1.47 | 2100 |

In fact, batch normalization cancels out the effect of the binarization/ternarization on the distribution of the gates and states during the training process. Moreover, batch normalization regulates the scale of binarized/ternarized weights using its parameter $\phi$ in addition to $\alpha$.

So far, we have only considered the forward computations. During the parameter update, we use full-precision weights since the parameter updates are small values. To update the full-precision weights, we use Eq. (1) to estimate its gradient since the binarization/ternarization functions are indifferentiable (See Algorithm 1 and its details in Appendix B). It is worth noting that using batch normalization makes the training process slower due to the additional computations required to perform Eq. (3).

## 5 EXPERIMENTAL RESULTS AND DISCUSSIONS

In this section, we evaluate the performance of the proposed LSTMs with binary/ternary weights on different temporal tasks to show the generality of our method. We defer hyperparameters and tasks details for each dataset to Appendix C due to the limited space.

### 5.1 CHARACTER-LEVEL LANGUAGE MODELING

For the character-level modeling, the goal is to predict the next character and the performance is evaluated on bits per character (BPC) where lower BPC is desirable. We conduct quantization experiments on Penn Treebank (Marcus et al. (1993)), War & Peace (Karpathy et al. (2015)) and Linux Kernel (Karpathy et al. (2015)) corpora. For Penn Treebank dataset, we use a similar LSTM model configuration and data preparation to Mikolov et al. (2012). For War & Peace and Linux Kernel datasets, we also follow the LSTM model configurations and settings in (Karpathy et al. (2015)). Table 1 summarizes the performance of our binarized/ternarized models compared to state-of-the-art quantization methods reported in literature. All the models reported in Table 1 use an LSTM layer with 1000, 512 and 512 units on a sequence length of 100 for the experiments on Penn Treebank (Marcus et al. (1993)), War & Peace (Karpathy et al. (2015)) and Linux Kernel (Karpathy et al. (2015)) corpora, respectively. The experimental results show that our model with binary/ternary weights outperforms all the existing quantized models in terms of prediction accuracy. Moreover, our ternarized model achieves the same BPC values on War & Peace and Penn Treebank datasets as the full-precision model (i.e., baseline) while requiring $32\times$ less memory footprint. It is worth mentioning the accuracy loss of our ternarized model over the full-precision baseline is small.

In order to evaluate the effectiveness of our method on a larger dataset for the character-level language modeling task, we use the Text8 dataset which was derived from Wikipedia. For this task, we use one

Table 2: Test character-level performance of our quantized models on the Text8 corpus.

| Model | Precision | Text8 | |
|---|---|---|---|
| | | Test (BPC) | Size (MByte) |
| LSTM (baseline) | Full-precision | 1.46 | 64.9 |
| LSTM with binary weights (ours) | Binary | 1.54 | 2.0 |
| LSTM with ternary weights (ours) | Ternary | **1.51** | 4.0 |
| BinaryConnect (Courbariaux et al. (2015)) | Binary | 2.45 | 2.0 |

LSTM layer of size 2000 and train it on sequences of length 180. We follow the data preparation approach and settings of Mikolov et al. (2012). The test results are reported in Table 2. While our models use recurrent binary or ternary weights during runtime, they achieve acceptable performance when compared to the full-precision models.

## 5.2 WORD-LEVEL LANGUAGE MODELING

Similar to the character-level language modeling, the main goal of word-level modeling is to predict the next word. However, this task deals with large vocabulary sizes, making the model quantization difficult. Xu et al. (2018) introduced a multi-bit quantization method, referred to as alternating method, as a first attempt to reduce the complexity of the LSTMs used for this task. However, the alternating method only managed to almost match its performance with its full-precision counterpart using 4 bits (i.e., $k = 4$). However, there is a huge gap in the performance between its quantized model with 2 bits and the full-precision one. To show the effectiveness of our method over the alternating method, we use a small LSTM of size 300 similar to (Xu et al. (2018)) for a fair comparison. We also examine the prediction accuracy of our method over the medium and large models introduced by Zaremba et al. (2014): the medium model contains an LSTM layer of size 650 and the large model contains two LSTM layers of size 1500. We also use the same settings described in (Mikolov et al. (2010)) to prepare and train our model. Table 3 summarizes the performance of our models in terms of perplexity. The experimental results show that our binarized/ternarized models outperform the alternating method using 2-bit quantization in terms of both perplexity and the memory size. Moreover, our medium-size model with binary weights also has a substantial improvement over the

Table 3: Test performance of the proposed LSTM with recurrent binary/ternary weights on the Penn Treebank (PTB) corpus.

| Model | Precision | Word-PTB | | |
|---|---|---|---|---|
| | | Test (Perplexity) | Size (KByte) | Operations (MOps) |
| Small LSTM (baseline) | Full-precision | 91.5 | 2880 | 1.4 |
| Small LSTM with binary weights (ours) | Binary | 92.2 | 90 | 1.4 |
| Small LSTM with ternary weights (ours) | Ternary | **90.7** | 180 | 1.4 |
| Small BinaryConnect LSTM (Courbariaux et al. (2015)) | Binary | 125.9 | 90 | 1.4 |
| Small Alternating LSTM (Xu et al. (2018)) | 2 bits | 103.1 | 180 | 2.9 |
| Small Alternating LSTM (Xu et al. (2018)) | 3 bits | 93.8 | 270 | 4.3 |
| Small Alternating LSTM (Xu et al. (2018)) | 4 bits | 91.4 | 360 | 5.8 |
| Medium LSTM (baseline) | Full-precision | 87.6 | 27040 | 13.5 |
| Medium LSTM with binary weights (ours) | Binary | 87.2 | 422 | 13.5 |
| Medium LSTM with ternary weights (ours) | Ternary | **86.1** | 845 | 13.5 |
| Medium BinaryConnect LSTM (Courbariaux et al. (2015)) | Binary | 108.4 | 422 | 13.5 |
| Large LSTM (baseline) | Full-precision | 78.5 | 144000 | 72 |
| Large LSTM with binary weights (ours) | Binary | 76.5 | 4500 | 72 |
| Large LSTM with ternary weights (ours) | Ternary | **76.3** | 9000 | 72 |
| Large BinaryConnect LSTM (Courbariaux et al. (2015)) | Binary | 128.5 | 4500 | 72 |

Table 4: Test accuracy of the proposed LSTM with recurrent binary/ternary weights on the pixel by pixel MNIST classification task.

| | | MNIST | | |
|---|---|---|---|---|
| Model | Precision | Test (%) | Size (KByte) | Operations (KOps) |
| LSTM (baseline) | Full-precision | 98.9 | 162 | 80.8 |
| LSTM with binary weights (ours) | Binary | 98.6 | 5 | 80.8 |
| LSTM with ternary weights (ours) | Ternary | **98.8** | 10 | 80.8 |
| BinaryConnect (Courbariaux et al. (2015)) | Binary | 68.3 | 5 | 80.8 |
| Alternating LSTM (Xu et al. (2018)) | 2 bits | 98.8 | 10 | 161.6 |

alternating method using 4-bit quantization. Finally, our models with recurrent binary and ternary weights yield a comparable performance compared to their full-precision counterparts.

## 5.3 SEQUENTIAL MNIST

We perform the MNIST classification task (Le et al. (2015)) by processing each image pixel at each time step. In this task, we process the pixels in scanline order. We train our models using an LSTM with 100 nodes, followed by a softmax classifier layer. Table 4 reports the test performance of our models with recurrent binary/ternary weights. While our binary model uses a lower bit precision and fewer operations for the recurrent computations compared to the alternating models, its loss of accuracy is small. On the other hand, our ternary model requires the same memory size and achieves the same accuracy as the alternating method while requiring $2\times$ fewer operations.

## 5.4 QUESTION ANSWERING

Hermann et al. (2015) recently introduced a challenging task that involves reading and comprehension of real news articles. More specifically, the main goal of this task is to answer questions about the context of the given article. To this end, they also introduced an architecture, called Attentive Reader, that exploits an attention mechanism to spot relevant information in the document. Attentive Reader uses two bidirectional LSTMs to encode the document and queries. To show the generality and effectiveness of our quantization method, we train Attentive Reader with our method to learn recurrent binary/ternary weights. We perform this task on the CNN corpus (Hermann et al. (2015)) by replicating Attentive Reader and using the setting described in (Hermann et al. (2015)). Table 5 shows the test accuracy of binarized/ternarized Attentive Reader. The simulation results show that our Attentive Reader with binary/ternary weights yields similar accuracy rate to its full-precision counterpart while requiring $32\times$ smaller memory footprint.

## 5.5 DISCUSSIONS

As discussed in Section 4, the training models ignoring the quantization loss fail to quantize the weights in LSTM while they perform well on CNNs and fully-connected networks. To address this problem, we proposed the use of batch normalization during the quantization process. To justify

Table 5: Test accuracy of Attentive Reader with recurrent binary/ternary weights on CNN question-answering task.

| | | CNN | |
|---|---|---|---|
| Model | Precision | Test (%) | Size (MByte) |
| Attentive Reader (baseline) | Full-precision | 59.81 | 7471 |
| Attentive Reader with binary weights (ours) | Binary | 59.22 | 233 |
| Attentive Reader with ternary weights (ours) | Ternary | **60.03** | 467 |
| BinaryConnect Attentive Reader (Courbariaux et al. (2015)) | Binary | 5.34 | 233 |

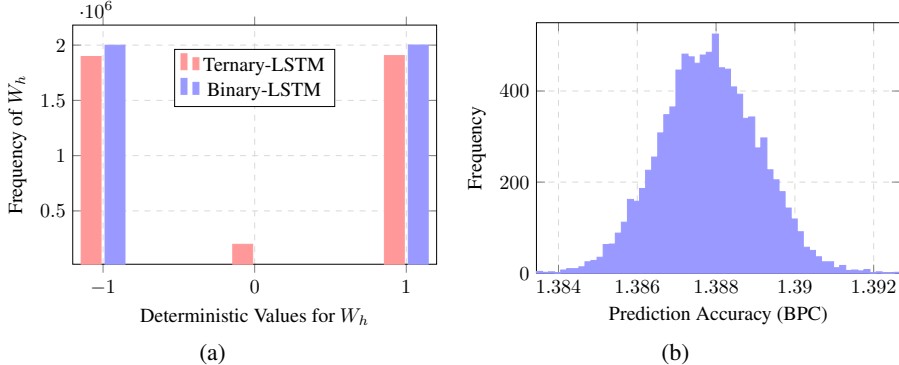

Figure 1: (a) Distribution of $W_h$ used in the LSTM layer on the Penn Treebank corpus. (b) Distribution of the prediction accuracy on the Penn Treebank corpus when using stochastic ternarization over 10000 samples.

the importance of such a decision, we have performed different experiments over a wide range of temporal tasks and compared the accuracy performance of our binarization/ternarization method with binaryconnect as a method that ignores the quantization loss. The experimental results showed that binaryconnect method fails to learn binary/ternary weights. On the other hand, our method not only learns recurrent binary/ternary weights but also outperforms all the existing quantization methods in literature. It is also worth mentioning that the models trained with our method achieve a comparable accuracy performance w.r.t. their full-precision counterpart.

Figure 1(a) shows a histogram of the binary/ternary weights of the LSTM layer used for character-level language modeling task on the Penn Treebank corpus. In fact, our model learns to use binary or ternary weights by steering the weights into the deterministic values of -1, 0 or 1. Despite the CNNs or fully-connected networks trained with binary/ternary weights that can use either real-valued or binary/ternary weights, the proposed LSTMs trained with binary/ternary can only perform the inference computations with binary/ternary weights. Moreover, the distribution of the weights is dominated by non-zero values for the model with ternary weights.

To show the effect of the probabilistic quantization on the prediction accuracy of temporal tasks, we adopted the ternarized network trained for the character-level language modeling tasks on the Penn Treebank corpus (see Section 5.1). We measured the prediction accuracy of this network on the test set over 10000 samples and reported the distribution of the prediction accuracy in Figure 1(b). Figure 1(b) shows that the variance imposed by the stochastic ternarization on the prediction accuracy is very small and can be ignored. It is worth mentioning that we also have observed a similar behavior for other temporal tasks used in this paper.

Figure 2 illustrates the learning curves and generalization of our method to longer sequences on the validation set of the Penn Treebank corpus. In fact, the proposed training algorithm also tries to retains the main features of using batch normalization, i.e., fast convergence and good generalization over long sequences. Figure 2(a) shows that our model converges faster than the full-precision LSTM for the first few epochs. After a certain point, the convergence rate of our method decreases, that prevents the model from early overfitting. Figure 2(b) also shows that our training method generalizes well over longer sequences than those seen during training. Similar to the full-precision baseline, our binary/ternary models learn to focus only on information relevant to the generation of the next target character. In fact, the prediction accuracy of our models improves as the sequence length increases since longer sequences provides more information from past for generation of the next target character.

While we have only applied our binarization/ternarization method on LSTMs, our method can be used to binarize/ternarize other recurrent architectures such as GRUs. To show the versatility of our method, we repeat the character-level language modeling task performed in Section 5.1 using GRUs on the Penn Treebank, War & Peace and Linux Kernel corpora. We also adopted the same network configurations and settings used in Section 5.1 for each of the aforementioned corpora. Table

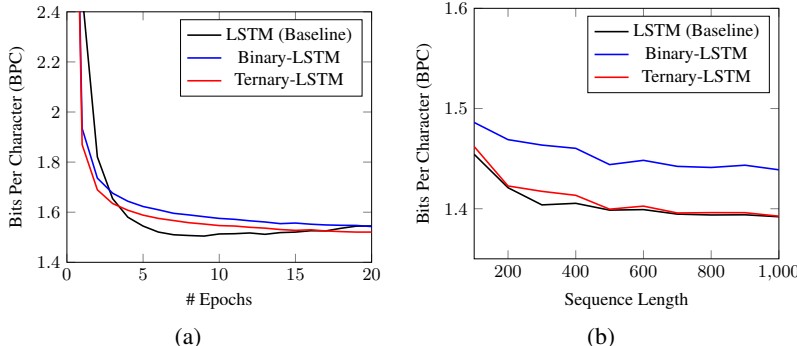

Figure 2: (a) The learning curve on the validation set of the Penn Treebank corpus. (b) Performance on the test set of the Penn Treebank corpus over longer sequences.

Table 6: Testing character-level BPC values of quantized GRU models and size of their weight matrices in terms of KByte.

| | | Linux Kernel | | War & Peace | | Penn Treebank | |
|---|---|---|---|---|---|---|---|
| Model | Precision | Test | Size | Test | Size | Test | Size |
| GRU (baseline) | Full-precision | 1.82 | 3772 | 1.75 | 3662 | 1.40 | 12612 |
| GRU with binary weights (ours) | Binary | 1.90 | 124 | 1.92 | 120 | 1.46 | 406 |
| GRU with ternary weights (ours) | Ternary | **1.81** | 241 | **1.82** | 235 | **1.41** | 799 |

6 summarizes the performance of our binarized/ternarized models. The simulation results show that our method can successfully binarize/ternarize the recurrent weights of GRUs.

As a final note, we have investigated the effect of using different batch sizes on the prediction accuracy of our binarized/ternarized models. To this end, we trained an LSTM of size 1000 over a sequence length of 100 and different batch sizes to perform the character-level language modeling task on the Penn Treebank corpus. Batch normalization cannot be used for the batch size of 1 as the output vector will be all zeros. Moreover, using a small batch size leads to a high variance when estimating the statistics of the unnormalized vector, and consequently a lower prediction accuracy than the baseline model without bath normalization, as shown in Figure 3. On the other hand, the prediction accuracy of our binarization/ternarization models improves as the batch size increases, while the prediction accuracy of the baseline model decreases.

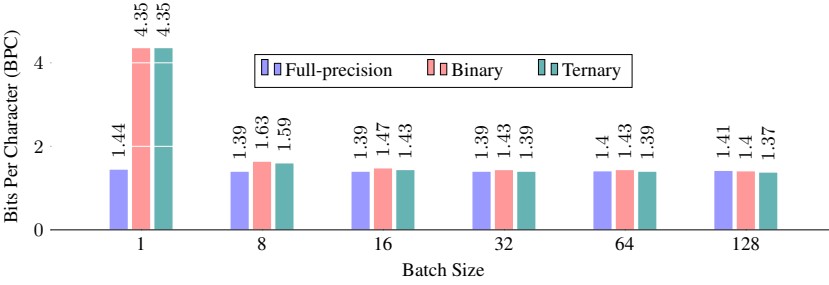

Figure 3: Effect of different batch sizes on the prediction accuracy of the character-level language modeling task on the Penn Treebank corpus.

Table 7: Implementation results of the proposed binary/ternary models vs full-precision models.

| | Low-Power | | | High-Speed | | |
|---|---|---|---|---|---|---|
| Model | Full-Precision | Binary | Ternary | Full-Precision | Binary | Ternary |
| # MAC Units | 100 | 100 | 100 | 100 | 1000 | 500 |
| Throughput (GOps/sec) | 80 | 80 | 80 | 80 | 800 | 400 |
| Silicon Area (mm$^2$) | 2.56 | 0.24 | 0.42 | 2.56 | 2.54 | 2.16 |
| Power (mW) | 336 | 37 | 61 | 336 | 347 | 302 |

## 6 HARDWARE IMPLEMENTATION

The introduced binarized/ternarized recurrent models can be exploited by various dataflows such as DaDianNao (Chen et al. (2014)) and TPU (Jouppi et al. (2017)). In order to evaluate the effectiveness of LSTMs with recurrent binary/ternary weights, we build our binary/ternary architecture over DaDianNao as a baseline which has proven to be the most efficient dataflow for DNNs with sigmoid/tanh functions. In fact, DaDianNao achieves a speedup of 656× and reduces the energy by 184× over a GPU (Chen et al. (2014)). Moreover, some hardware techniques can be adopted on top of DaDianNao to further speed up the computations. For instance, Zhang et al. (2016) showed that ineffectual computations of zero-valued weights can be skipped to improve the run-time performance of DaDianNao. In DaDianNao, a DRAM is used to store all the weights/activations and provide the required memory bandwidth for each multiply-accumulate (MAC) unit. For evaluation purposes, we consider two different application-specific integrated circuit (ASIC) architectures implementing Eq. (2): low-power implementation and high-speed inference engine. We build these two architectures based on the aforementioned dataflow. For the low-power implementation, we use 100 MAC units. We also use a 12-bit fixed-point representation for both weights and activations of the full-precision model as a baseline architecture. As a result, 12-bit multipliers are required to perform the recurrent computations. Note that using the 12-bit fixed-point representation for weights and activations guarantees no prediction accuracy loss in the full-precision models. For the LSTMs with recurrent binary/ternary weights, a 12-bit fixed-point representation is only used for activations and multipliers in the MAC units are replaced with low-cost multiplexers. Similarly, using 12-bit fixed-point representation for activations guarantees no prediction accuracy loss in the introduced binary/ternary models. We implemented our low-power inference engine for both the full-precision and binary/ternary-precision models in TSMC 65-nm CMOS technology. The synthesis results excluding the implementation cost of the DRAM are summarized in Table 7. They show that using recurrent binary/ternary weights results in up to 9× lower power and 10.6× lower silicon area compared to the baseline when performing the inference computations at 400 MHz.

For the high-speed design, we consider the same silicon area and power consumption for both the full-precision and binary/ternary-precision models. Since the MAC units of the binary/ternary-precision model require less silicon area and power consumption as a result of using multiplexers instead of multipliers, we can instantiate up to 10× more MAC units, resulting in up to 10× speedup compared to the full-precision model (see Table 7). It is also worth noting that the models using recurrent binary/ternary weights also require up to 12× less memory bandwidth than the full-precision models. More details on the proposed architecture are provided in Appendix D.

## 7 CONCLUSION

In this paper, we introduced a method that learns recurrent binary/ternary weights and eliminates most of the full-precision multiplications of the recurrent computations during the inference. We showed that the proposed training method generalizes well over long sequences and across a wide range of temporal tasks such as word/character language modeling and pixel by pixel classification tasks. We also showed that learning recurrent binary/ternary weights brings a major benefit to custom hardware implementations by replacing full-precision multipliers with hardware-friendly multiplexers and reducing the memory bandwidth. For this purpose, we introduced two ASIC implementations: low-power and high-throughput implementations. The former architecture can save up to 9× power consumption and the latter speeds up the recurrent computations by a factor of 10.

ACKNOWLEDGMENT

The authors would like to thank Loren Lugosch for his helpful suggestions.

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

APPENDIX A

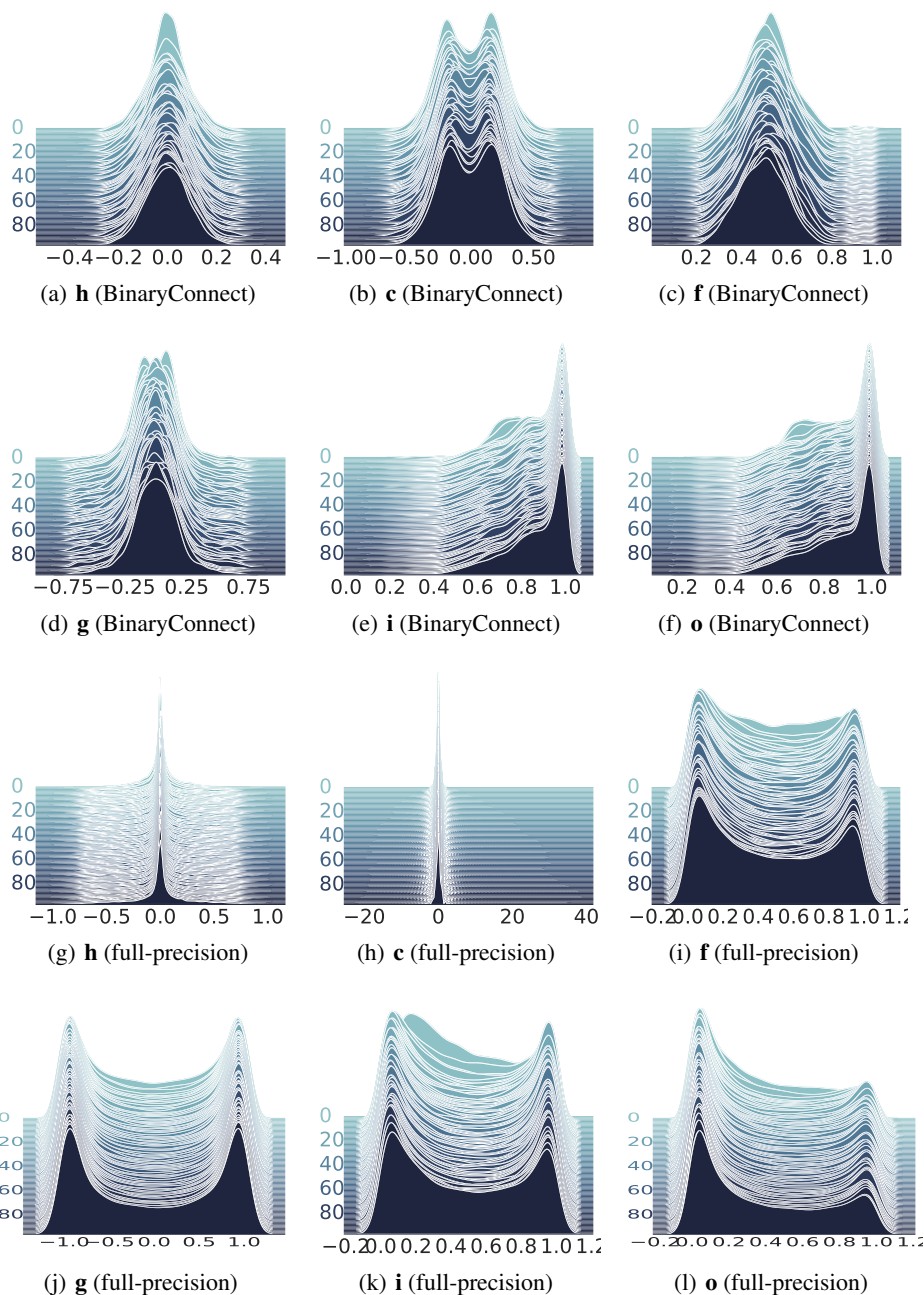

Figure 4: Probability density of states/gates for the BinaryConnect LSTM compared to its full-precision counterpart on the Penn Treebank character-level modeling task. Both models were trained for 50 epochs. The vertical axis denotes the time steps.

Figure 4 shows the probability density of the gates and hidden states of the BinaryConnect LSTM and its full-precision counterpart both trained with 1000 units and a sequence length of 100 on Penn Treebank corpus Marcus et al. (1993) for 50 epochs. The probability density curves show that the gates in the binarized LSTM fail to control the flow of information. More specifically, the input gate **i** and the output gate **o** tend to let all information through, the gate **g** tends to block all information, and the forget gate $f$ cannot decide to let which information through.

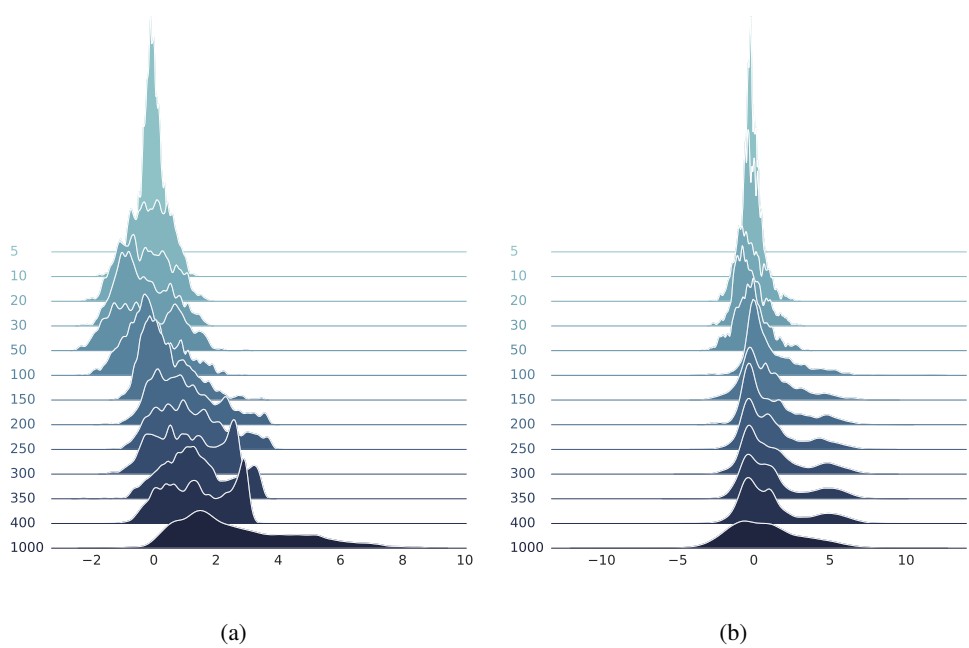

(a)                                                        (b)

Figure 5: Probability density of $\mathbf{i}^p$ for (a) the BinaryConnect LSTM compared to (b) its full-precision counterpart for a single time step at different training iterations on the Penn Treebank character-level modeling task. The vertical axis denotes the training iterations.

Figure 5 illustrates the probability density of $\mathbf{i}^p$ at different iterations during the training process. The simulation results show a shift in probability density of $\mathbf{i}^p$ during the training process, resulting in all positive values for $\mathbf{i}^p$ and values centered around 1 for the input gate $\mathbf{i}$. In fact, the binarization process changes the probability density of the gates and hidden states during the training phase.

APPENDIX B

Learning recurrent binary/ternary weights are performed in two steps: forward propagation and backward propagation.

**Forward Propagation**: A key point to learn recurrent binary/ternary weights is to batch-normalize the result of each vector-matrix multiplication with binary/ternary recurrent weights during the forward propagation. More precisely, we first binarize/ternarize the recurrent weights. Afterwards, the unit activations are computed while using the recurrent binarized/ternarized weights for each time step and recurrent layer. The unit activations are then normalized during the forward propagation.

**Backward Propagation**: During the backward propagation, the gradient with respects to each parameter of each layer is computed. Then, the updates for the parameters are obtained using a learning rule. During the parameter update, we use full-precision weights since the parameter updates are small values. More specifically, the recurrent weights are only binarized/ternarized during the forward propagation. Algorithm 1 summarizes the training method that learns recurrent binary/ternary weights. It is worth noting that batch normalizing the state unit $\mathbf{c}$ can optionally be used to better control over its relative contribution in the model.

---

**Algorithm 1:** Training with recurrent binary/ternary weights. $\ell$ is the cross entropy loss function. B/T() specifies the binarization/ternarization function. The batch normalization transform is also denoted by $\text{BN}(\cdot; \phi, \gamma)$. $L$ and $T$ are also the number of LSTM layers and time steps, respectively.

---

**Data:** Full-precision LSTM parameters $\mathbf{W}_{fh}, \mathbf{W}_{ih}, \mathbf{W}_{oh}, \mathbf{W}_{gh}, \mathbf{W}_{fx}, \mathbf{W}_{ix}, \mathbf{W}_{ox}, \mathbf{W}_{gx}, \mathbf{b}_f, \mathbf{b}_i,$ $\mathbf{b}_o$ and $\mathbf{b}_g$ for each layer. Batch normalization parameters for hidden-to-hidden and input-to-hidden states. The classifier parameters $\mathbf{W}_s$ and $\mathbf{b}_s$. Input data $\mathbf{x}^1$, its corresponding targets $\mathbf{y}$ for each minibatch.

1 **1. Forward Computations**
2 **for** $l = 1$ *to* $L$ **do**
3   $\mathbf{W}^l_{fh_{B/T}} \leftarrow \text{B/T}(\mathbf{W}^l_{fh}), \mathbf{W}^l_{ih_{B/T}} \leftarrow \text{B/T}(\mathbf{W}^l_{ih})$
4   $\mathbf{W}^l_{oh_{B/T}} \leftarrow \text{B/T}(\mathbf{W}^l_{oh}), \mathbf{W}^l_{gh_{B/T}} \leftarrow \text{B/T}(\mathbf{W}^l_{gh})$
5   $\mathbf{W}^l_{fx_{B/T}} \leftarrow \text{B/T}(\mathbf{W}^l_{fx}), \mathbf{W}^l_{ix_{B/T}} \leftarrow \text{B/T}(\mathbf{W}^l_{ix})$
6   $\mathbf{W}^l_{ox_{B/T}} \leftarrow \text{B/T}(\mathbf{W}^l_{ox}), \mathbf{W}^l_{gx_{B/T}} \leftarrow \text{B/T}(\mathbf{W}^l_{gx})$
7   **for** $t = 1$ *to* $T$ **do**
8    $\mathbf{f}^l_t = \sigma\Big(\text{BN}(\mathbf{W}^l_{fh_{B/T}}\mathbf{h}^l_{t-1}; \phi^l_{fh}, 0) + \text{BN}(\mathbf{W}^l_{fx_{B/T}}\mathbf{x}^l_t; \phi^l_{fx}, 0) + \mathbf{b}^l_f\Big)$
9    $\mathbf{i}^l_t = \sigma\Big(\text{BN}(\mathbf{W}^l_{ih_{B/T}}\mathbf{h}^l_{t-1}; \phi^l_{ih}, 0) + \text{BN}(\mathbf{W}^l_{ix_{B/T}}\mathbf{x}^l_t; \phi^l_{ix}, 0) + \mathbf{b}^l_i\Big)$
10    $\mathbf{o}^l_t = \sigma\Big(\text{BN}(\mathbf{W}^l_{oh_{B/T}}\mathbf{h}^l_{t-1}; \phi^l_{oh}, 0) + \text{BN}(\mathbf{W}^l_{ox_{B/T}}\mathbf{x}^l_t; \phi^l_{ox}, 0) + \mathbf{b}^l_o\Big)$
11    $\mathbf{g}^l_t = \tanh\Big(\text{BN}(\mathbf{W}^l_{gh_{B/T}}\mathbf{h}^l_{t-1}; \phi^l_{gh}, 0) + \text{BN}(\mathbf{W}^l_{gx_{B/T}}\mathbf{x}^l_t; \phi^l_{gx}, 0) + \mathbf{b}^l_g\Big)$
12    $\mathbf{c}^l_t = \mathbf{f}^l_t \odot \mathbf{c}^l_{t-1} + \mathbf{i}^l_t \odot \mathbf{g}^l_t$
13    $\mathbf{h}^l_t = \mathbf{o}^l_t \odot \tanh\big(\text{BN}(\mathbf{c}^l_t; \phi^l_c, \gamma^l_c)\big)$
14   **end**
15   $\mathbf{x}^{l+1} = \mathbf{h}^l$
16 **end**
17 $\hat{\mathbf{y}} = \text{softmax}(\mathbf{W}_s\mathbf{h}^L + \mathbf{b}_s)$
18 **2. Backward Computations**
19 Compute the loss function $\ell$ knowing $\hat{\mathbf{y}}$ and $\mathbf{y}$
20 Obtain the updates $\Delta\mathbf{W}_s$ and $\Delta\mathbf{b}_s$ by computing $\dfrac{\partial\ell}{\partial\mathbf{W}_s}$ and $\dfrac{\partial\ell}{\partial\mathbf{b}_s}$, respectively

21 **for** $l = 1$ *to* $L$ **do**
22   Obtain the updates $\Delta\mathbf{W}^l_{fh}, \Delta\mathbf{W}^l_{ih}, \Delta\mathbf{W}^l_{oh}, \Delta\mathbf{W}^l_{gh}, \Delta\mathbf{W}^l_{fx}, \Delta\mathbf{W}^l_{ix}, \Delta\mathbf{W}^l_{ox}$ and $\Delta\mathbf{W}^l_{gx}$ by
   computing $\dfrac{\partial\ell}{\partial\mathbf{W}^l_{fh_{B/T}}}, \dfrac{\partial\ell}{\partial\mathbf{W}^l_{ih_{B/T}}}, \dfrac{\partial\ell}{\partial\mathbf{W}^l_{oh_{B/T}}}, \dfrac{\partial\ell}{\partial\mathbf{W}^l_{gh_{B/T}}}, \dfrac{\partial\ell}{\partial\mathbf{W}^l_{fx_{B/T}}}, \dfrac{\partial\ell}{\partial\mathbf{W}^l_{ix_{B/T}}},$
   $\dfrac{\partial\ell}{\partial\mathbf{W}^l_{ox_{B/T}}}$ and $\dfrac{\partial\ell}{\partial\mathbf{W}^l_{gx_{B/T}}}$, respectively
23   Obtain the updates $\Delta\mathbf{b}^l_f, \Delta\mathbf{b}^l_i, \Delta\mathbf{b}^l_o, \Delta\mathbf{b}^l_g, \Delta\mathbf{h}^l_0$ and $\Delta\mathbf{c}^l_0$ by computing $\dfrac{\partial\ell}{\partial\mathbf{b}^l_f}, \dfrac{\partial\ell}{\partial\mathbf{b}^l_i}, \dfrac{\partial\ell}{\partial\mathbf{b}^l_o},$
   $\dfrac{\partial\ell}{\partial\mathbf{b}^l_g}, \dfrac{\partial\ell}{\partial\mathbf{h}^l_0}$ and $\dfrac{\partial\ell}{\partial\mathbf{c}^l_0}$, respectively
24   Obtain the updates $\Delta\phi^l_{fh}, \Delta\phi^l_{ih}, \Delta\phi^l_{oh}, \Delta\phi^l_{gh}, \Delta\phi^l_{fx}, \Delta\phi^l_{ix}, \Delta\phi^l_{ox}, \Delta\phi^l_{gx}, \Delta\phi^l_c$ and $\Delta\gamma^l_c$
   by computing $\dfrac{\partial\ell}{\partial\phi^l_{fh}}, \dfrac{\partial\ell}{\partial\phi^l_{ih}}, \dfrac{\partial\ell}{\partial\phi^l_{oh}}, \dfrac{\partial\ell}{\partial\phi^l_{gh}}, \dfrac{\partial\ell}{\partial\phi^l_{fx}}, \dfrac{\partial\ell}{\partial\phi^l_{ix}}, \dfrac{\partial\ell}{\partial\phi^l_{gx}}, \dfrac{\partial\ell}{\partial\phi^l_{ox}}, \dfrac{\partial\ell}{\partial\phi^l_c}$ and $\dfrac{\partial\ell}{\partial\gamma^l_c}$,
   respectively
25 **end**
26 Update the network parameters using their updates

---

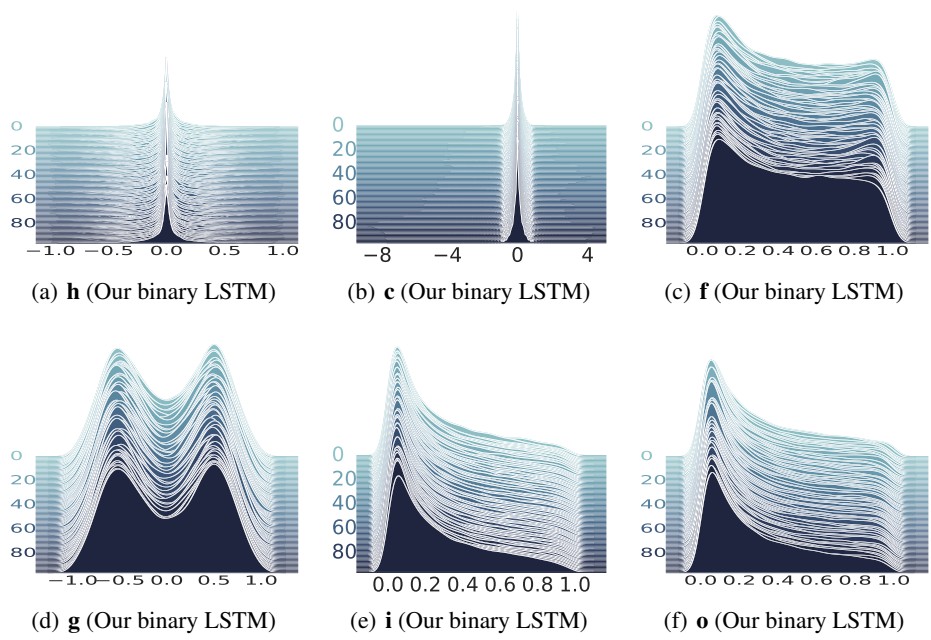

Figure 6: Probability density of states/gates for our binarized LSTM on the Penn Treebank character-level modeling task. The model was trained for 50 epochs. The vertical axis denotes the time steps.

## APPENDIX C

### C.1 CHARACTER-LEVEL LANGUAGE MODELING

**Penn Treebank**: Similar to Mikolov et al. (2012), we split the Penn Treebank corpus into 5017k, 393k and 442k training, validation and test characters, respectively. For this task, we use an LSTM with 1000 units followed by a softmax classifier. The cross entropy loss is minimized on minibatches of size 64 while using ADAM learning rule. We use a learning rate of 0.002. We also use the training sequence length of size 100. Figure 6 depicts the probability density of the states/gates of our binarized model trained on the Penn Treebank corpus. While the probability density of our model is different from its full-precision counterpart (see Figure 4), it shows that the gates can control the flow of information.

**Linux Kernel and Leo Tolstoy's War & Peace**: Linux Kernel and Leo Tolstoy's War and Peace corpora consist of 6,206,996 and 3,258,246 characters and have a vocabulary size of 101 and 87, respectively. We split these two datasets similar to Karpathy et al. (2015). We use one LSTM layer of size 512 followed by a softmax classifier layer. We use an exponentially decaying learning rate initialized with 0.002. ADAM learning rule is also used as the update rule.

**Text8**: This dataset has the vocabulary size of 27 and consists of 100M characters. Following the data preparation approach of Mikolov et al. (2012), we split the data into training, validation and test sets as 90M, 5M and 5M characters, respectively. For this task, we use one LSTM layer of size 2000 and train it on sequences of length 180 with minibatches of size 128. The learning rate of 0.001 is used and the update rule is determined by ADAM.

### C.2 WORD-LEVEL LANGUAGE MODELING

**Penn Treebank**: Similar to Mikolov et al. (2010), we split the Penn Treebank corpus with a 10K size vocabulary, resulting in 929K training, 73K validation, and 82K test tokens. We start the training with a learning rate of 20. We then divide it by 4 every time we see an increase in the validation perplexity value. The model is trained with the word sequence length of 35 and the dropout probability of 0.5,

0.65 and 0.65 for the small, medium and large models, respectively. Stochastic gradient descent is also used to train our model while the gradient norm is clipped at 0.25.

### C.3 SEQUENTIAL MNIST

**MNIST**: MNIST dataset contains 60000 gray-scale images (50000 for training and 10000 for testing), falling into 10 classes. For this task, we process the pixels in scanline order: each image pixel is processed at each time step similar to Le et al. (2015). We train our models using an LSTM with 100 nodes, a softmax classifier layer and ADAM step rule with learning rate of 0.001.

### C.4 QUESTION ANSWERING

**CNN**: For this task, we split the data similar to Hermann et al. (2015). We adopt Attentive Reader architecture to perform this task. We train the model using bidirectional LSTM with unit size of 256. We also use minibatches of size 128 and ADAM learning rule. We use an exponentially decaying learning rate initialized with 0.003.

### APPENDIX D

We implemented our binary/ternary architecture in VHDL and synthesized via Cadence Genus Synthesis Solution using TSMC 65nm GP CMOS technology. Figure 7 shows the latency of the proposed binary/ternary architecture for each time step and temporal task when performing the vector-matrix multiplications on binary/ternary weights. The simulation results show that performing the computations on binary and ternary weights can speed up the computations by factors of $10\times$ and $5\times$ compared to the full-precision models.

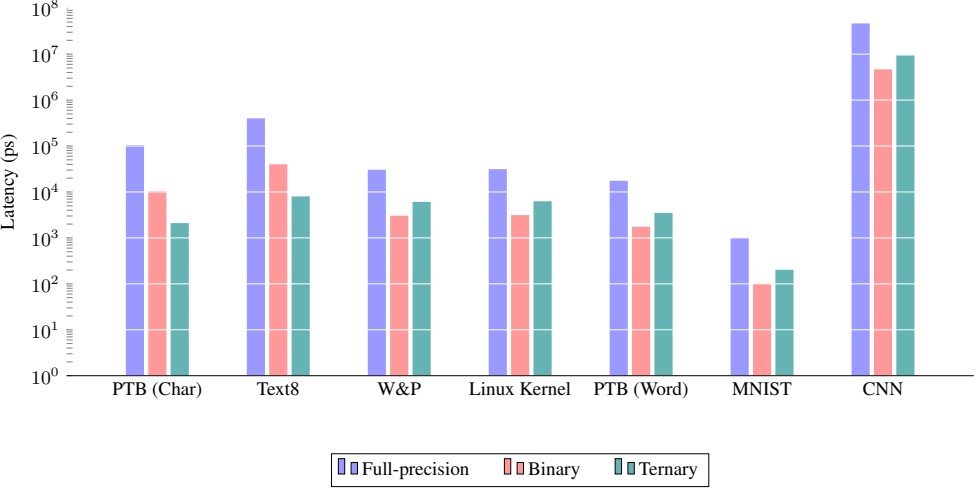

Figure 7: Latency of the proposed accelerator over full-precision, binary and ternary models.

