# OpenReview forum: "Learning Recurrent Binary/Ternary Weights"
_ICLR.cc/2019/Conference_

### Official Review · AnonReviewer3 · 2018-10-24
**Batch normalization for RNNs with binary and ternary weights**

**Rating:** 7
**Confidence:** 4

**Review:**

* Summary
This paper proposes batch normalization for learning RNNs with binary or ternary weights instead of full-precision weights. Experiments are carried out on character-level and word-level language modeling, as well as sequential MNIST and question answering.


* Strengths
- I liked the variety of tasks used evaluations (sequential MNIST, language modeling, question answering).
- Encouraging results on specialized hardware implementation.


* Weaknesses
- Using batch normalization on existing binarization/ternarization techniques is a bit of an incremental contribution.
- All test perplexities for word-level language models in table 3 underperform compared to current vanilla LSTMs for that task (see Table 4 in https://arxiv.org/pdf/1707.05589.pdf), suggesting that the baseline LSTM used in this paper is not strong enough.
- Results on question answering are not convincing -- BinaryConnect has the same size while achieving substantially higher accuracy (94.66% vs 40.78%). This is nowhere discussed and the paper's major claims "binaryconnect method fails" and "our method [...] outperforms all the existing quantization methods" seem unfounded (Section 5.5).
- In the introduction, I am lacking a distinction between improvements w.r.t. training vs inference time. As far as I understand, quantization methods only help at reducing memory footprint or computation time during inference/test but not during training. This should be clarified.
- In the introduction on page 2 is argued that the proposed method "eliminates the need for multiplications" -- I do not see how this is possible. Maybe what you meant is that it eliminates the need for full-precision multiplications by replacing them with multiplications with binary/ternary matrices?
- The notation is quite confusing. For starters, in Section 2 you mention "a fixed scaling factor A" and I would encourage you to indicate scalars by lower-case letters, vectors by boldface lower-case letters and matrices by boldface upper-case letters. Moreover, it is unclear when calculations are approximate. For instance, in Eq. 1 I believe you need to replace "=" with "\approx". Likewise for the equation in the next to last line on page 2. Lastly, while Eq. 2 seems to be a common way to write down LSTM equations, it is abusive notation.


* Minor Comments
- Abstract: What is ASIC? It is not referenced in Section 6.
- Introduction: What is the justification for calling RNNs over-parameterized? This seems to depend on the task.
- Introduction; contributions: Here, I would like to see a distinction between gains during training vs test time.
- Section 3.2 comes out of nowhere. You might want to already mention why are introducing batch normalization at this point.
- The boldfacing in Table 1, 2 and 3 is misleading. I understand this is done to highlight the proposed method, but I think commonly boldfacing is used to highlight the best results.
- Figure 2b. What is your hypothesis why BPC actually goes down the longer the sequence is?
- Algorithm 1, line 14: Using the cross-entropy is a specific choice dependent on the task. My understanding is your approach can work with any differentiable downstream loss?

---

> ### Author Response · Authors · 2018-11-26
> **To Reviewer #3 (3/3)**
>
> ---------------------------------------------------------
>
> Minor Comments:
>
> ---------------------------------------------------------
>
> Reviewer comment: Abstract: What is ASIC? It is not referenced in Section 6.
>
> Our response: Application-Specific Integrated Circuit (ASIC) is a common term used as an integrated circuit customized for a particular use. We have now defined this abbreviation in Section 6 of the revised manuscript.
>
> ---------------------------------------------------------
>
> Reviewer comment: Introduction: What is the justification for calling RNNs over-parameterized? This seems to depend on the task.
>
> Our response: We agree with the reviewer that calling all RNNs over-parameterized is not precise, as it highly depends on the task and dimensions of inputs/outputs/state vectors of RNNs. Based on your comment, we have changed the sentence to “..., RNNs are typically over-parameterized ...” in the introduction section of the revised manuscript. It is worth mentioning that it has been shown in literature that most networks’ parameters can be pruned or quantized without any performance degradation, suggesting that neural networks are typically over-parameterized.
>
> ---------------------------------------------------------
>
> Reviewer comment: Introduction; contributions: Here, I would like to see a distinction between gains during training vs test time.
>
> Our response: Based on your comment, we have clearly mentioned in the revised manuscript (see the first bullet point of Section 1 and the last sentence of Section 4) that using binarized/ternarized weights is only beneficial for inference.
>
> ---------------------------------------------------------
>
> Reviewer comment: Section 3.2 comes out of nowhere. You might want to already mention why are introducing batch normalization at this point.
>
> Our response: We completely agree with the reviewer on this point. Based on your comment, we have merged Section 3.2 of the submitted version with Section 4. We now introduce batch normalization right after explaining why we are motivated to use it in the revised manuscript.
>
> ---------------------------------------------------------
>
> Reviewer comment: The boldfacing in Table 1, 2 and 3 is misleading. I understand this is done to highlight the proposed method, but I think commonly boldfacing is used to highlight the best results.
>
> Our response: Based on your comment, we have only highlighted the best results in all the tables of the revised manuscript.
>
> ---------------------------------------------------------
>
> Reviewer comment: Figure 2b. What is your hypothesis why BPC actually goes down the longer the sequence is?
>
> Our response: We believe that the models learn to focus only on information relevant to the generation of the next target character. The prediction accuracy of the models improves as the sequence length increases since longer sequences provide more information from the past to generate the next target character. We have added the above discussion to Section 5.5 of the revised manuscript.
>
> ---------------------------------------------------------
>
> Reviewer comment: Algorithm 1, line 14: Using the cross-entropy is a specific choice dependent on the task. My understanding is your approach can work with any differentiable downstream loss?
>
> Our response: Thank you for raising this. We have also used our method with other loss functions, and it is not limited to only cross entropy. We have addressed the issue in the revised manuscript.

---

> > ### Comment · AnonReviewer3 · 2018-11-28
> > **Convincing Revision**
> >
> > I think the authors for the in-depth response and revision. I increased my score from 5 to 7.

---

> ### Author Response · Authors · 2018-11-26
> **To Reviewer #3 (2/3)**
>
> Reviewer comment: In the introduction, I am lacking a distinction between improvements w.r.t. training vs inference time. As far as I understand, quantization methods only help at reducing memory footprint or computation time during inference/test but not during training. This should be clarified.
>
> Our response: Based on your comment, we have clearly mentioned that our method only helps at reducing memory footprint and computation time during inference in the introduction section of the revised manuscript.
>
> ---------------------------------------------------------
>
> Reviewer comment: In the introduction on page 2 is argued that the proposed method "eliminates the need for multiplications" -- I do not see how this is possible. Maybe what you meant is that it eliminates the need for full-precision multiplications by replacing them with multiplications with binary/ternary matrices?
>
> Our response: Thank you for raising this. Yes, we meant that it eliminates the need for full-precision multiplication by replacing them with multiplications with binary/ternary matrices. It is worth mentioning that a multiplication with one binary or ternary multiplicand is implemented with a very simple hardware circuit, much smaller and more power efficient than a full-precision or multi-bit multiplier. We have revised the statement in the manuscript.
>
> ---------------------------------------------------------
>
> Reviewer comment: The notation is quite confusing. For starters, in Section 2 you mention "a fixed scaling factor A" and I would encourage you to indicate scalars by lower-case letters, vectors by boldface lower-case letters and matrices by boldface upper-case letters. Moreover, it is unclear when calculations are approximate. For instance, in Eq. 1 I believe you need to replace "=" with "\approx". Likewise for the equation in the next to last line on page 2. Lastly, while Eq. 2 seems to be a common way to write down LSTM equations, it is abusive notation.
>
> Our response: Based on your comment, all the notations have been addressed in the revised manuscript.

---

> ### Author Response · Authors · 2018-11-26
> **To Reviewer #3 (1/3)**
>
> We sincerely thank the reviewer for careful reading of our manuscript and many insightful comments and suggestions towards improving our paper. Below we respond to each comment of yours in detail.
>
> ---------------------------------------------------------
>
> Major Comments (Weaknesses):
>
> ---------------------------------------------------------
>
> Reviewer comment: Using batch normalization on existing binarization/ternarization techniques is a bit of an incremental contribution.
>
> Our response: We agree with the reviewer that the idea may sound simple, but we believe it can be counted as a strength of our paper due to its effectiveness for the binarization/ternarization process. While most of the existing binarization/ternarization methods are specialized for a specific temporal task, we have shown that our method can perform equally good over different temporal tasks and outperform the existing quantization methods in terms of prediction accuracy. We believe our method has paved the way for hardware designers to exploit LSTMs with binarized/ternarized weights which require less implementation cost for embedded systems.
>
> ---------------------------------------------------------
>
> Reviewer comment: All test perplexities for word-level language models in table 3 underperform compared to current vanilla LSTMs for that task (see Table 4 in https://arxiv.org/pdf/1707.05589.pdf), suggesting that the baseline LSTM used in this paper is not strong enough.
>
> Our response: Based on the reviewer’s comment, we adopted the large LSTM model (Zaremba et al. (2014)) in Table 4 of the mentioned paper as a baseline and performed our binarization/ternarization method for this baseline. The simulation results are reported in Table 3 of the revised manuscript and show that our binarized and ternarized models outperform the baseline model in terms of perplexity. More precisely, our binarized and ternarized models respectively achieve perplexity values of 76.5 and 75.3 while the perplexity value of the baseline model is 78.5.
>
> ---------------------------------------------------------
>
> Reviewer comment: Results on question answering are not convincing -- BinaryConnect has the same size while achieving substantially higher accuracy (94.66% vs 40.78%). This is nowhere discussed and the paper's major claims "binaryconnect method fails" and "our method [...] outperforms all the existing quantization methods" seem unfounded (Section 5.5).
>
> Our response: Thank you for the comment. In the submitted manuscript, we had reported the test error rate as opposed to accuracy of different methods for the question answering task in Table 5. However, we agree with the reviewer that since Table 4 reported accuracy and Table 5 reported error rate, there was an inconsistency between Table 4 and Table 5 that could cause confusion. Based on the reviewer’s comment, we have now summarized the results of Table 5 in terms of accuracy rate in the revised manuscript to make them consistent with the results of the sequential image classification task in Table 4. According to Table 5 of the revised manuscript, BinaryConnect completely fails to learn the question answering task while our method yields a similar accuracy rate to its full-precision baseline. We showed that our method can binarize/ternarize weights of a more realistic RNN application.
>
> ---------------------------------------------------------

---

### Official Review · AnonReviewer1 · 2018-11-02
**Simple but useful method, substantial experiments**

**Rating:** 8
**Confidence:** 3

**Review:**

This work proposes a method for reducing memory requirements in RNN models via binary / ternary quantisation. The authors argue that binarising RNNs is due to a covariate shift, and address it with stochastic quantised weights and batch normalisation.
The proposed RNN is tested on 6 sequence modelling tasks/datasets and shows drastic memory improvements compared to full-precision RNNs, with almost no loss in test performance.
Based on the more efficient RNN cell, the authors furthermore describe a more efficient hardware implementation, compared to an implementation of the full-precision RNN.

The core message I took away from this work is: “One can get away with stochastic binarised weights in a forward pass by compensating for it with batch normalisation”.

Strengths:
- substantial number of experiments (6 datasets), different domains
- surprisingly simple methodological fix
- substantial literature review
- it has been argued that char-level / pixel-level RNNs present somewhat artificial tasks — even better that the authors test for a more realistic RNN application (Reading Comprehension) with an actually previously published model.

Weaknesses:
- little understanding is provided into _why_ covariance shift occurs/ why batch normalisation is so useful. The method works, but the authors could elaborate more on this, given that this is the core argument motivating the chosen method.
- some statements are too bold/vague , e.g. page 3: “a binary/ternary model that can perform all temporal tasks”
- unclear: by adapting a probabilistic formulation / sampling quantised weights, some variance is introduced. Does it matter for predictions (which should now also be stochastic)? How large is this variance? Even if negligible, it is not obvious and should be addressed.


Other Questions / Comments
-  How dependent is the method on the batch size chosen? This is in particular relevant as smaller batches might yield poor empirical estimates for mean/var. What happens at batch size 1? Are predictions of poorer for smaller batches?
- Section 2, second line — detail: case w_{i,j}=0 is not covered
- equation (5): total probability mass does not add up to 1
- a direct comparison with models from previous work would have been interesting, where these previous methods also rely on batch normalisation
- as I understand, the main contribution is in the inference (forward pass), not in training. It is somewhat misleading when the authors speak about “the proposed training algorithm” or “we introduced a training algorithm”
- unclear: last sentence before section 6.

---

> ### Author Response · Authors · 2018-11-26
> **To Reviewer #1 (2/2)**
>
> ---------------------------------------------------------
>
> Other Questions / Comments
>
> ---------------------------------------------------------
>
> Reviewer comment: How dependent is the method on the batch size chosen? This is in particular relevant as smaller batches might yield poor empirical estimates for mean/var. What happens at batch size 1? Are predictions of poorer for smaller batches?
>
> Our response: Based on your comment, we have investigated the effect of using different batch sizes on the prediction accuracy of our binarized/ternarized models (see Section 5.5 of the revised manuscript). To this end, we trained an LSTM of size 1000 over a sequence length of 100 and different batch sizes to perform the character-level language modeling task on the Penn Treebank corpus. The simulation results show that batch normalization cannot be used with batch size of 1, as the output vector will be all zeros. Moreover, using batch sizes slightly larger than 1 lead to a high variance in the estimations of the statistics of the unnormalized vector, resulting in a lower prediction accuracy than the baseline model (without batch normalization) as shown in Figure 3 of the revised manuscript. On the other hand, the prediction accuracy of our binarized/ternarized models improves as the batch size increases while the prediction accuracy of the baseline model decreases.
>
> ---------------------------------------------------------
>
> Reviewer comment: Section 2, second line — detail: case w_{i,j}=0 is not covered
>
> Our response: Thank you for raising this. We have fixed the typo in the revised manuscript.
>
> ---------------------------------------------------------
>
> Reviewer comment: equation (5): total probability mass does not add up to 1
>
> Our response: For the stochastic ternarization process, we sample from [0, 1] interval depending on the weight sign. In case of having a positive sign, the probability of getting +1 is equal to the absolute value of the normalized weight and the probability of getting 0 is 1-P(w = 1) which adds up to 1. Similarly, in case of having a negative sign, the probability of getting -1 is equal to the absolute value of the normalized weight and the probability of getting 0 is 1-P(w = -1) which also adds up to 1.
>
> ---------------------------------------------------------
>
> Reviewer comment: a direct comparison with models from previous work would have been interesting, where these previous methods also rely on batch normalisation
>
> Our response: Unfortunately, we could not find any other methods that rely on batch normalization to the best of our knowledge. However, we tried our best to compare our method with other existing quantization methods.
>
> ---------------------------------------------------------
>
> Reviewer comment: as I understand, the main contribution is in the inference (forward pass), not in training. It is somewhat misleading when the authors speak about “the proposed training algorithm” or “we introduced a training algorithm”
>
> Our response: We completely agree with the reviewer. We have revised the misleading statements in the manuscript based on your comment.
>
> ---------------------------------------------------------
>
> Reviewer comment: unclear: last sentence before section 6.
>
> Our response: Thank you for raising this. We have rephrased the sentence in the revised manuscript.
>
> ---------------------------------------------------------

---

> ### Author Response · Authors · 2018-11-26
> **To Reviewer #1 (1/2)**
>
> We sincerely thank the reviewer for careful reading of our manuscript and many insightful comments and suggestions towards improving our paper. Below we respond to each comment of yours in detail.
>
> ---------------------------------------------------------
>
> Major Comments (Weaknesses):
>
> ---------------------------------------------------------
>
> Reviewer comment: little understanding is provided into _why_ covariance shift occurs/ why batch normalisation is so useful. The method works, but the authors could elaborate more on this, given that this is the core argument motivating the chosen method.
>
> Our response: The main motivation of using batch normalization for the binarization/ternarization process relies on an observation of distribution of an LSTM gates/states trained with the BinaryConnect method. More precisely, we first trained an LSTM using the BinaryConnect for 50 epochs and illustrated the distribution of gates/states (see Fig. 4 of the revised manuscript). Comparing the distribution curves of the BinaryConnect model with its full-precision counterpart shows that the BinaryConnect method makes LSTM ineffective. In fact, the LSTM gates that are supposed to control the flow of information fail to function properly. For instance, the output gate o and the input gate i tend to let all information through while these gates in the full-precision model behaves differently. To explore the cause of this problem, we performed the second experiment: we measured the distribution of the input gate i before its non-linear function applied during different training iterations. We observed that the binarization process changes the distribution and pushes it towards positive values (see Fig. 5 in the revised manuscript) during the training process. Motivated from these observations, we decided to use batch normalization as it provides more robustness to the network. Based on the reviewer’s comment, we have added the above discussion to Section 4 of the revised manuscript.
>
> ---------------------------------------------------------
>
> Reviewer comment: some statements are too bold/vague , e.g. page 3: “a binary/ternary model that can perform all temporal tasks”
>
> Our response: Thank you for raising this issue. We have revised the bold/vague statements in the manuscript.
>
> ---------------------------------------------------------
>
> Reviewer comment: unclear: by adapting a probabilistic formulation / sampling quantised weights, some variance is introduced. Does it matter for predictions (which should now also be stochastic)? How large is this variance? Even if negligible, it is not obvious and should be addressed.
>
> Our response: In fact, the variance that is introduced by the stochastic binarization/ternarization process on the prediction accuracy is very small. For instance, we measured the distribution of the prediction accuracy on the Penn Treebank corpus when using the stochastic ternarization process over 10000 samples as shown in Fig. 1 (b) of the revised manuscript. This curve shows that the variance imposed by the stochastic process is negligible on the prediction accuracy. Based on the reviewer’s comment, we have added the above discussion to Section 5.5 of the revised manuscript.

---

### Official Review · AnonReviewer4 · 2018-11-13
**Achieving binary/ternary LSTMs using batch normalization within recurrent layers**

**Rating:** 6
**Confidence:** 3

**Review:**

The paper proposes a method to achieve binary and ternary quantization for recurrent networks. The key contribution is applying batch normalization to both input matrix vector and hidden matrix vector products within recurrent layers in order to preserve accuracy. The authors demonstrate accuracy benefits on a variety of datasets including language modeling (character and word level), MNIST sequence, and question answering. A hardware implementation based on DaDianNao is provided as well.

Strengths
- The authors propose a relatively simple and easy to understand methodology for achieving aggressive binary and ternary quantization.
- The authors present compelling accuracy benefits on a range of datasets.

Weaknesses / Questions
- While the application of batch normalization demonstrates good results, having more compelling results on why covariate shift is such a problem in LSTMs would be helpful. Is this methodology applicable to other recurrent layers like RNNs and GRUs?
- Does applying batch normalization across layer boundaries or at the end of each time-step help? This may incur lower overhead during inference and training time compared to applying batch normalization to the output of each matrix vector product (inputs and hidden-states).
- Does training with batch-normalization add additional complexity to the training process? I imagine current DL framework do not efficiently parallelize applying batch normalization on both input and hidden matrix vector products.
- It would be nice to have more intuition on what execution time overheads batch-normalization applies during inference on a CPU or GPU. That is, without a hardware accelerator what are the run-time costs, if any.
- The hardware implementation could have much more detail. First, where are the area and power savings coming from. It would be nice to have a breakdown of on-chip SRAM for weights and activations vs. required DRAM memory. Similarly having a breakdown of power in terms of on-chip memory, off-chip memory, and compute would be helpful.
- The hardware accelerator baseline assumes a 12-bit weight and activation quantization. Is this the best that can be achieved without sacrificing accuracy compared to floating point representation? Does adding batch normalization to intermediate matrix-vector products increase the required bit width for activations to preserve accuracy?

Other comments
- Preceding section 3.2 there no real discussion on batch normalization and covariate shift which are central to the work’s contribution. It would be nice to include this in the introduction to guide the reader.
- It is unclear why DaDianNao was chosen as the baseline hardware implementation as opposed to other hardware accelerator implementations such as TPU like dataflows or the open-source NVDLA.

---

> ### Author Response · Authors · 2018-11-26
> **To Reviewer #4 (3/3)**
>
> Reviewer comment: The hardware accelerator baseline assumes a 12-bit weight and activation quantization. Is this the best that can be achieved without sacrificing accuracy compared to floating point representation? Does adding batch normalization to intermediate matrix-vector products increase the required bit width for activations to preserve accuracy?
>
> Our response: According to our simulation results, both the baseline and the proposed models require 12 bits for activations without incurring any accuracy degradation. The weights of the baseline model also require 12 bits for a fixed-point representation without incurring any performance degradation. Similar results have also been reported in ESE paper (see https://dl.acm.org/citation.cfm?id=3021745). Based on your comment, we have added the above discussion to Section 6 of the revised manuscript.
>
> ---------------------------------------------------------
>
> Other comments
>
> ---------------------------------------------------------
>
> Reviewer comment: Preceding section 3.2 there no real discussion on batch normalization and covariate shift which are central to the work’s contribution. It would be nice to include this in the introduction to guide the reader.
>
> Our response: We completely agree with the reviewer. We have now merged Section 3.2 with Section 4 in the revised manuscript to make a more coherent statement.
>
> ---------------------------------------------------------
>
> Reviewer comment: It is unclear why DaDianNao was chosen as the baseline hardware implementation as opposed to other hardware accelerator implementations such as TPU like dataflows or the open-source NVDLA.
>
> Our response: We agree with the reviewers that TPU and NVDLA are among the best accelerators reported to-date. However, we believe that DaDianNao is also one of the best accelerators and the reason for that is twofold. First, DaDianNao is designed for energy-efficiency: it can process neural computations ~656x faster and is ~184x more energy efficient than GPUs (see https://ieeexplore.ieee.org/document/7480791). Second, some hardware techniques can be adopted on top of DaDianNao to further speed up the computations. For instance, in  Cambricon-X paper (see https://ieeexplore.ieee.org/document/7783723), it was shown that sparsity among both activations and weights can be exploited on top of the DaDianNao’s dataflow to skip the noncontributory computations with zeros and speed up the process. Similarly, Cnvlutin’s paper (see https://ieeexplore.ieee.org/document/7551378) uses the DaDianNao’s architecture to skip the noncontributory computations of zero-valued activations. We believe that similar techniques can be also exploited to skip the noncontributory computations of zero-valued weights of ternarized RNNs as a future work. Based on the reviewer’s comment, we have added the above discussion to the revised manuscript.

---

> ### Author Response · Authors · 2018-11-26
> **To Reviewer #4 (2/3)**
>
> Reviewer comment: Does training with batch-normalization add additional complexity to the training process? I imagine current DL framework do not efficiently parallelize applying batch normalization on both input and hidden matrix vector products.
>
> Our response: Yes, it adds additional complexity and makes the training process slightly slow. However, since we target embedded devices requiring real-time inference process, the additional complexity in the training process is a worthwhile tradeoff when considering the gain that batch normalization provides for hardware implementations (i.e., having binary/ternary weights which requires less hardware cost). Based on the reviewer’s comment, we have added a discussion stating that batch normalization introduces additional complexity to the training process.
>
> ---------------------------------------------------------
>
> Reviewer comment: It would be nice to have more intuition on what execution time overheads batch-normalization applies during inference on a CPU or GPU. That is, without a hardware accelerator what are the run-time costs, if any.
>
> Our response: Batch normalization by itself makes the inference computations slower (by a factor of ~1.3 in our simulation) on a CPU or GPU platform. On the other hand, binarized/ternarized weights can be exploited to speed up the computations. For instance, XNOR-Net paper (https://arxiv.org/abs/1603.05279) has shown that using binarized weights can speed up the computations by a factor of 2 on a CPU platform. Moreover, using binarized/ternarized weights saves memory and reduces the memory access and consequently power consumption. Unfortunately, due to the limited time that we had for the rebuttal period, we could not measure the inference time of our method on a CPU or GPU platform.
>
> ---------------------------------------------------------
>
> Reviewer comment: The hardware implementation could have much more detail. First, where are the area and power savings coming from. It would be nice to have a breakdown of on-chip SRAM for weights and activations vs. required DRAM memory. Similarly having a breakdown of power in terms of on-chip memory, off-chip memory, and compute would be helpful.
>
> Our response: In fact, the power and area savings come from replacing the full-precision multipliers with binary/ternary multipliers (i.e., multiplexers). Table 6 only reports the implementation results of the computational core excluding the DRAM that was used to store all the weights and activations. Depending on the target application, the size of DRAM could be different. It is worth mentioning that the main computational core builds on a large array of Multiply-and-Accumulate units working in parallel. The intermediate results are stored into registers, and the computed activations are written into the DRAM. The breakdown of storage required to store the weights for each task is reported in Table 1 to 5 in Section 5. Based on your comment, we have added the above discussion to the paper.
>
> ---------------------------------------------------------

---

> ### Author Response · Authors · 2018-11-26
> **To Reviewer #4 (1/3)**
>
> We sincerely thank the reviewer for careful reading of our manuscript and many insightful comments and suggestions towards improving our paper. Below we respond to each comment of yours in detail.
>
> ---------------------------------------------------------
>
> Major Comments (Weaknesses):
>
> ---------------------------------------------------------
>
> Reviewer comment: While the application of batch normalization demonstrates good results, having more compelling results on why covariate shift is such a problem in LSTMs would be helpful. Is this methodology applicable to other recurrent layers like RNNs and GRUs?
>
> Our response: The main motivation of using batch normalization for the binarization/ternarization process relies on an observation of distribution of an LSTM gates/states trained with the BinaryConnect method. More precisely, we first trained an LSTM using the BinaryConnect for 50 epochs and illustrated the distribution of gates/states (see Fig. 4 of the revised manuscript). Comparing the distribution curves of the BinaryConnect model with its full-precision counterpart shows that the BinaryConnect method makes LSTM ineffective. In fact, the LSTM gates that are supposed to control the flow of information fail to function properly. For instance, the output gate o and the input gate i tend to let all information through while these gates in the full-precision model behaves differently. To explore the cause of this problem, we performed the second experiment: we measured the distribution of the input gate i before its non-linear function applied during different training iterations. We observed that the binarization process changes the distribution and pushes it towards positive values (see Fig. 5 in the revised manuscript) during the training process. Motivated from these observations, we decided to use batch normalization as it provides more robustness to the network. Based on the reviewer’s comment, we have added the above discussion to Section 4 of the revised manuscript.
>
> To also show the applicability of our method to GRUs, we repeated the character-level language modeling task performed in Section 5.1 while using GRUs instead of LSTMs on the Penn Treebank, War & Peace and Linux Kernel corpora. We also adopted the same network configurations and settings used in Section 5.1 for each of the aforementioned corpora. Table 6 summarizes the performance of our binarized/ternarized models. The simulation results show that our method can successfully binarize/ternarize the recurrent weights of GRUs. Based on the reviewer’s comment, we have added the above discussion to Section 5.5 of the revised manuscript.
>
> ---------------------------------------------------------
>
> Reviewer comment: Does applying batch normalization across layer boundaries or at the end of each time-step help? This may incur lower overhead during inference and training time compared to applying batch normalization to the output of each matrix vector product (inputs and hidden-states).
>
> Our response: Since the immediate impact of the binarization/ternarization process is on the value of gates in LSTM, it works the best when batch normalization is applied right after the vector-matrix multiplications.
>
> ---------------------------------------------------------

---

### Public Comment · (anonymous) · 2018-12-01
**clarification for experiment settings**

Thank the authors for the simple and effective methodology for  binary and ternary quantization in LSTMs. However, the experiment settings are not very explicitly stated in the paper. I tried out the released code online and have some questions about the experiment settings, which could be important for fair evaluation for the efficacy of the proposed method.  Could the authors clarify a bit about inconsistency or the implicit part in the experiment settings?

1. what is the optimizer used for word-level language modeling on PTB data set?
The submitted paper does not mention what kind of optimizer is used. After checking the code, I found that it is vanilla SGD, but scaled by a "norm_factor", which is the squared 2 norm of the gradient.  Could the authors clarify it in the paper why this particular scaling parameter is chosen? Moreover, if the particular scaling is crucial to the performance, for fair comparison, the baseline methods compared should also use this kind of optimization.

2. What is the sequence length used for character-level language modeling on text8 dataset? The paper says it is 180, but the released code shows it is 200. Which one is correct? and does this cause a significant difference in the final performance? What sequence length is used for the compared baseline methods?

3. This paper used sequence length 35 for the word-level language modeling task on Penn Treebank. However, the Alternating LSTM (Xu et al. (2018)) uses 30. It is known that usually the larger sequence length, the better performance. For this aspect, the comparison may not be fair.

---

> ### Author Response · Authors · 2018-12-05
> **Re: clarification for experiment settings**
>
> We sincerely thank the reader for careful reading of our manuscript and code. Below we respond to each comment of yours in detail.
> ----------------------------------------------------
> Comment: 1. what is the optimizer used for word-level language modeling on PTB data set? The submitted paper does not mention what kind of optimizer is used. After checking the code, I found that it is vanilla SGD, but scaled by a "norm_factor", which is the squared 2 norm of the gradient. Could the authors clarify it in the paper why this particular scaling parameter is chosen? Moreover, if the particular scaling is crucial to the performance, for fair comparison, the baseline methods compared should also use this kind of optimization.
>
> Our response: Thank you for the comment. Due to the limited space, we had to defer the details of training settings to Appendix C in the paper. In Appendix C.2, we have mentioned that we only use vanilla SGD while clipping the gradient norm at 0.25. We also used this setting not only for our models but also for the baseline models reported in Table 3 for a fair comparison. Moreover, the Alternating LSTM (Xu et al. (2018)) was trained with the same setting (please see Section 5 of the Alternating paper). Since all the models reported in Table 3 were trained with the same setting, we believe that we have constructed a fair comparison. It is also worth mentioning that while we used SGD with the norm clipping method for this task, our method is not limited to only this setting. In fact, we have trained our models using different optimizers such as Adam and achieved comparable perplexity values. For instance, our medium LSTM with ternary weights trained with the Adam optimizer yields a perplexity value of 90. However, we agree that vanilla SGD with the norm clipping method works better, which explains why other works in literature (such as Xu et al. (2018) and Zaremba et al. (2014)) use this setting.
> ----------------------------------------------------
> Comment: 2. What is the sequence length used for character-level language modeling on text8 dataset? The paper says it is 180, but the released code shows it is 200. Which one is correct? and does this cause a significant difference in the final performance? What sequence length is used for the compared baseline methods?
>
> Our response: Thank you for the comment. All the results reported in Table 2 were obtained using the sequence length of 180. However, since we have been using the code for our other works, we simply forgot to restore the original settings that we used for this paper. Based on your comment, we have updated the code with the original settings that we used to obtain the results reported in Table 2.
> ----------------------------------------------------
> Comment: 3. This paper used sequence length 35 for the word-level language modeling task on Penn Treebank. However, the Alternating LSTM (Xu et al. (2018)) uses 30. It is known that usually the larger sequence length, the better performance. For this aspect, the comparison may not be fair.
>
> Our response: Thank you for the comment. For this task, we adopted the model introduced by Zaremba et al. (2014) (please see https://arxiv.org/pdf/1409.2329.pdf) as our baseline which uses the sequence length of 35. Using the sequence length of 35 is a common choice for this task.  For example, all the models reported in Table 4 in https://arxiv.org/pdf/1707.05589.pdf use the same sequence length. We also believe that learning longer-term dependencies is more desirable and challenging when using LSTMs. As a result, we followed the same trend.
>
> In Table 3, we showed that our ternary models match their performance with their baseline while there is a large gap between the 2-bit Alternating LSTM model and its baseline (please see Table 1 in (Xu et al. (2018))). Regardless of the sequence length, our model can match its performance with the baseline while the Alternating model fails to do so when using 2 bits for the representation of weights. Additionally, based on your comment, we have also trained our small LSTM models with the sequence length of 30, and obtained perplexity values of 92.4 and 90.7 for the small binary and ternary LSTM models, respectively. In fact, the obtained results for the sequence length of 30 (i.e., 92.4 for the binary model and 90.7 for the ternary model) are very similar to the results obtained for the sequence length of 35 (i.e., 92.2 for the binary model and 90.7 for the ternary model).

---

### Meta-Review · Area_Chair1 · 2018-12-14
**Simple but strong method**

**Confidence:** 5
**Recommendation:** Accept (Poster)

**Metareview:**

This work proposes a simple but useful way to train RNN with binary / ternary weights for improving memory and power efficiency. The paper presented a sequence of experiments on various benchmarks and demonstrated significant improvement on memory size  with only minor decrease of accuracy. Authors' rebuttal addressed the reviewers' concern nicely.